# Role of Fatty Acids β-Oxidation in the Metabolic Interactions Between Organs

**DOI:** 10.3390/ijms252312740

**Published:** 2024-11-27

**Authors:** Alexander V. Panov, Vladimir I. Mayorov, Sergey I. Dikalov

**Affiliations:** 1Department of Biomedical Sciences, Mercer University School of Medicine, Macon, GA 31201, USA; Mayorov_vi@mercer.edu; 2Vanderbilt University Medical Center, Nashville, TN 37232, USA; sergey.dikalov@vanderbilt.edu

**Keywords:** aerobic glycolysis, beta-oxidation, gluconeogenesis, fatty acids, lactate, mitochondria, metabolism, respirasome

## Abstract

In recent decades, several discoveries have been made that force us to reconsider old ideas about mitochondria and energy metabolism in the light of these discoveries. In this review, we discuss metabolic interaction between various organs, the metabolic significance of the primary substrates and their metabolic pathways, namely aerobic glycolysis, lactate shuttling, and fatty acids β-oxidation. We rely on the new ideas about the supramolecular structure of the mitochondrial respiratory chain (respirasome), the necessity of supporting substrates for fatty acids β-oxidation, and the reverse electron transfer via succinate dehydrogenase during β-oxidation. We conclude that ATP production during fatty acid β-oxidation has its upper limits and thus cannot support high energy demands alone. Meanwhile, β-oxidation creates conditions that significantly accelerate the cycle: glucose-aerobic glycolysis-lactate-gluconeogenesis-glucose. Therefore, glycolytic ATP production becomes an important energy source in high energy demand. In addition, lactate serves as a mitochondrial substrate after converting to pyruvate + H^+^ by the mitochondrial lactate dehydrogenase. All coupled metabolic pathways are irreversible, and the enzymes are organized into multienzyme structures.

## 1. Introduction

“Tell me what you eat, and I will tell you who you are”   Jean Anthelme Brillat-Savarin

The above statement was made in 1826 by Jean Anthelme Brillat-Savarin, a French politician, and gastronome, and became a proverb: “You are what you eat”. But how true is this proverb? Finding the answer to this seemingly simple question is not easy since, upon deeper reflection, the question has many sides. There are so many publications on various aspects of diets, nutrition, and metabolism that deciding what is essential and fundamental and what is secondary or simply nonsense becomes very confusing. Different diets and methods of limiting food consumption have gained wide popularity. Some diets are very extreme, such as “vegan” diets. A vegetarian eats no animal flesh, such as meat, poultry, or fish. A vegan is a stricter vegetarian who also avoids consuming dairy, eggs, and any other ingredients derived from animals. Of note, the “popularity” of vegetarianism and veganism in some countries results not from the devotion to a healthy lifestyle but due to the population’s poverty, hot climate, and lack of modern food storage, particularly during the previous centuries. This article clarifies the confusion of facts, ideas, and opinions about human metabolism. Before we start, we must discuss the state of our knowledge about the role of mitochondria in metabolic processes in the human body. For several objective and subjective reasons, our knowledge and understanding of the mechanisms and physiology of metabolic processes has significant gaps or, sometimes, is significantly distorted. For example, 60 years ago, physiologists, in experiments with isolated organ perfusion, found that the heart and kidneys obtain up to 80–90% of their energy through β-oxidation of long-chain fatty acids (LCFAs) [1,2]. However, we still do not know why isolated mitochondria from these organs do not oxidize the acyl-carnitines of LCFAs at high rates. In the following sections, we will discuss the potential causes of errors and misconceptions and try to evaluate existing methods of influencing human metabolism.

## 2. The Main Features of Metabolism in Mammals

Biologically, humans belong to the class Mammalia, subgroup Theria, and the next subgroup Eutheria, which comprises placental mammals. The placental mammals are divided into herbivores, carnivores, and omnivores, according to their primary food source. Evolutionarily, humans are omnivores, although, in our time, some people tend to change their eating habits by shifting to their herbivorous or carnivorous side. Despite the principal differences in food composition between herbivores and carnivores, the structure and functions of the main organs, such as the heart, kidneys, and skeletal muscles, are very similar. Correspondingly, the metabolic features of these organs are also identical for all mammals, except the liver, which is responsible for converting very different foodstuffs of the herbivores and carnivores into similar proteins and lipids of the main organs. Several functions of the central nervous system and the central metabolic events in some organs and functional systems are distributed between different types and groups of cells, similarly in many mammals. Therefore, not the composition of the food but the structure and functions determine the organ’s metabolic features. Because plant fiber is difficult to digest and assimilate, unlike meat, there are significant differences between herbivores and predators in the size and function of the liver and digestive system. Omnivores, including humans, share some metabolic and digestive features with herbivores and carnivore animals. In addition to the structural and functional similarities of many organs, a common feature in all mammals is an extensive dependence on the microbial fermentation of polysaccharides in the hindgut, where large amounts of short-chain fatty acids (SCFA) are produced [3].

It is important to note that mitochondria play a central role in cellular metabolism both for energy production and synthetic functions. Therefore, most nutrients will go through mitochondria regardless of the diet. Meanwhile, mitochondrial plasticity adjusts its structure and function to specific organs and physiological conditions. In the following sections, our attention will be directed to human metabolism in general and the metabolic features of the main organs: the heart, brain, kidneys, skeletal muscles, and liver. Except for the liver, metabolism in human organs has much in common with other mammals, especially omnivores. This allows us to use animal experimental data to understand human metabolism better.

## 3. Types of Experiments and Information Regarding Metabolism Obtained from Animals

Whole animals, predominantly rats and mice, have been used to study the effects of various diets and starvation on animal functions and tissue composition. In the 20th century, one of the most important discoveries was that restricted diets (reduced to 30% from ad libitum food consumption) could significantly expand the life span of experimental animals [4,5].

### 3.1. Importance of the Substrate Combinations for the In Vitro Experiments with Isolated Mitochondria

In the perfusion experiments with the isolated organs (heart, liver, and kidney), researchers discovered that ATP production in the heart and kidney is predominantly supported by the β-oxidation of long-chain fatty acids [1,2]. Because mitochondria are the center of major metabolic events in the body and for most organs and tissues, oxidative phosphorylation is the primary source of ATP; therefore, for several decades, isolated mitochondria from different organs have been used for studies on energy metabolism and mitochondrial functions. Numerous fundamental discoveries have been made regarding electron transport, the mechanism of ATP production, the transportation of cations and anions, and so on. However, many physiological aspects of energy metabolism in mitochondria and their role in the body’s metabolic processes remain unknown or misunderstood. For a long time, we have encountered significant discrepancies between experimental physiology, which uses a whole organ, and in vitro studies with isolated mitochondria. In vivo and the isolated organ perfusion experiments, mitochondria metabolize several substrates simultaneously. In contrast, in the in vitro experiments, researchers usually provided mitochondria with only a single substrate, sometimes with malate, which has a supporting role but is not oxidized. Therefore, we observe poor acyl-carnitine oxidation in all mitochondria isolated from different organs. Only in 2018, it was shown that active β-oxidation of long-chain fatty acids requires the simultaneous presence of other mitochondrial metabolites, succinate, glutamate, or pyruvate [6,7,8]. For the isolated synaptic brain mitochondria, maximal rates of oxidative phosphorylation were found with the mixture of pyruvate + glutamate + malate [9].

### 3.2. Endogenous Inhibition of Succinate Dehydrogenase (SDH)

Activation of fatty acids β-oxidation, which occurs in the presence of supporting substrates [6], is associated with the reversal of the succinate dehydrogenase (SDH) reaction when the electrons flow from the membrane’s pool of CoQH_2_ to the tricarboxylic acid cycle (discussed in [10]). Thus, SDH (complex II) plays a critical role in fatty acid beta-oxidation by reversing electron flow and preventing catabolism of mitochondrial metabolites. Therefore, it is important to discuss another phenomenon observed in the in vitro experiments with succinate as a sole substrate: the endogenous inhibition of SDH caused by oxaloacetate. This inhibition is relieved by adding glutamate (see Figure 1 and Figure 2), indicating the importance of mixtures of mitochondrial metabolites.

Freshly isolated liver mitochondria possess a relatively large matrix volume and thus contain a relatively large amount of intramitochondrial substrates, which can support state 3 respiration for more than 10 min. Therefore, the isolated liver mitochondria oxidize succinate at high rates in the resting and phosphorylating metabolic states. The isolated heart mitochondria have a much smaller matrix volume than the liver mitochondria and fewer intramitochondrial intrinsic substrates. Therefore, they show the regular rate of state 4 respiration but fail to activate succinate oxidation upon adding ADP (Figure 2). Adding glutamate or pyruvate instantly activates oxidative phosphorylation (Figure 1 and Figure 2).

Synaptic brain mitochondria are tiny and do not contain endogenous substrates. No oxygen is consumed when synaptic mitochondria are added to the incubation medium without substrate. Therefore, brain mitochondria require a constant supply of substrates from astroglia. The optimum substrate mixture for synaptic mitochondria, which provides high respiration rates, is pyruvate + glutamate + malate [9]. However, synaptic mitochondria oxidize long-chain fatty acids (LCFAs) in the presence of pyruvate, glutamate, or succinate [12]. This challenges the old myth that the brain consumes only glucose to satisfy its energy requirements.

Figure 3 shows a comprehensive study of the respiratory activity of rat brain synaptic mitochondria oxidizing various substrates and their mixtures [12]. Traditionally, in the in vitro experiments with isolated mitochondria, researchers used only one substrate, sometimes with malate, which is a very poor substrate but promotes the transport and oxidation of some substrates. Figure 3A,B show respiratory activities in metabolic state 4 (before the addition of ADP) and metabolic state 3 (after activation of oxidative phosphorylation by ADP). The respiratory control ratio (RCR = v State 3/v state 4) shows the degree of oxygen consumption stimulation upon adding ADP. We can see that both with glutamate and pyruvate, oxygen consumption rates are low, and the addition of ADP increases respiratory rates by 11 and 7.5-fold, correspondingly. However, the rates of oxidative phosphorylation with these substrates are relatively low. We have provided evidence that the mixture of glutamate + pyruvate + malate doubles the rate of oxidative phosphorylation in synaptic mitochondria [9]. Figure 3C shows succinate, when used alone, is a poor substrate because adding ADP does not activate oxidative phosphorylation. Figure 3D shows that the substrate mixture of succinate + glutamate + pyruvate significantly increases the rate of oxidative phosphorylation. This supports the idea that, unlike the in vitro system, in vivo, mitochondria oxidize a mixture of substrates [9]. Figure 3E shows that palmitoyl carnitine is a poor substrate in the absence of supporting substrates, whereas in the presence of pyruvate (Figure 3F), glutamate (Figure 3G), or succinate (Figure 3H), the rates of oxidative phosphorylation increase significantly. Interestingly, when used alone, succinate (Figure 3C) and palmitoyl-carnitine (Figure 3E) are poor substrates for brain mitochondria. However, when both substrates are added together (Figure 3H), we observed the highest oxygen consumption rates in states 4 and 3.

Another phenomenon is related to respiration coupling and mitochondrial superoxide production. For example, during the in vitro oxidation of succinate alone (Figure 3C) and in the presence of glutamate + pyruvate (Figure 3D) or palmitoyl-carnitine (Figure 3H), the rates of oxygen consumption are increased more than threefold in the resting metabolic state (absence of ADP), when compared with the brain or mitochondria oxidizing “classical” substrates glutamate + malate (Figure 3A) and pyruvate + malate (Figure 3B), and this was associated with a severalfold increase in superoxide production [12]. It is unclear if the association between resting mitochondrial oxygen consumption and superoxide production is preserved in vivo, and the pathophysiological role of this phenomenon needs further investigation. This question arose after it was discovered that β-oxidation involves a very complex functional superstructure of many mitochondrial enzymatic complexes and that mitochondrial transhydrogenase transfers the excess of mitochondrial-reducing equivalents, created by the reversed electron flow, from NADH + H^+^ to the cytoplasmic NADPH + H^+^ [10].

## 4. Methodological Flaws That Negatively Impacted Metabolism Research

Our analysis suggests that several methodological mistakes have delayed or misled the progress in understanding many aspects of mitochondrial physiology and metabolism. Here, we mention only some of the principal problems. The first problem was excluding female rats and mice from the studies. Scientists believed that data obtained on females would be less reliable than using males because females have periods of estrus, pregnancy, and hormonal seasonal shifts. Decades later, it was discovered that many aspects of metabolism have fundamental gender differences. Females of all mammals produce less reactive oxygen species, age slower, and live longer [13,14], possibly because they oxidize fatty acids differently from the males (reviewed in [15]).

The second problem is due to overreliance on isolated liver mitochondria, which were the main research object for several decades and thus became a standard for comparison with mitochondria from other organs. Now we know that liver mitochondria have several unique features that make them principally different from the heart and brain mitochondria [16]. The problem is that oxidative phosphorylation in liver mitochondria strongly depends on the metabolic state and time after feeding [16]. The details are described below. Here is enough to mention that researchers usually isolate liver mitochondria from overnight-starved rats and mice to obtain more reproducible data. Because mice are much smaller than rats, they have a higher rate of metabolism, so the degree of response to overnight starvation is much stronger than in rats. The liver from the starved animals oxidizes fatty acids and has active gluconeogenesis. For this reason, the pyruvate dehydrogenase complex is inhibited, and most other metabolites of the tricarboxylic acids cycle become non-physiological substrates. Unfortunately, isolated mitochondria from all organs in the in vitro system oxidize the carnitine esters of the long-chain fatty acids very poorly, including the brain mitochondria (Figure 3E). For this reason, researchers rarely used acylcarnitines as a substrate. Therefore, researchers started using succinate + rotenone or glutamate + malate as substrates for most experiments with liver mitochondria, underestimating the role of fatty acids in metabolism.

When researchers began isolating mitochondria from the heart and brain, they found that brain mitochondria from rats and mice often do not oxidize succinate even in state 4 (resting respiration in the absence of added ADP). Heart mitochondria oxidize succinate in state 4 but do not stimulate respiration upon adding ADP (Figure 2). However, when brain and heart mitochondria were isolated in the presence of 0.1% defatted bovine serum albumin (BSA), the mitochondria oxidized succinate well in metabolic states 4 and 3. After adding an uncoupler, respiration became inhibited. However, when mitochondria were supplemented with succinate in the presence of rotenone, mitochondria respired in all metabolic states [16]. Researchers thought that inhibition of succinate dehydrogenase was an in vitro artifact, and thus, succinate + rotenone became a universal substrate for mitochondria from all organs [16]. In the following sections, we will discuss the critical role of the succinate-dependent reverse electron transport for mitochondrial and cell metabolism, which is excluded by rotenone, and that endogenous inhibition of succinate dehydrogenase activity plays essential roles in controlling energy metabolism and modulation of superoxide production in the brain and the heart.

## 5. The Metabolic Properties of Liver Mitochondria Play Unique and Crucial Roles in the Liver’s Metabolic States Distinct from Those of the Heart or Brain Mitochondria

According to the British Liver Trust [17], the liver has over 500 functions. The most important of them are (a) processing digested food from the intestine; (b) maintaining metabolic (substrate) homeostasis for other organs and tissues by controlling levels of fats, amino acids, and glucose in the blood; (c) neutralizing and destroying drugs and toxins; (d) making enzymes and proteins, which are responsible for most chemical reactions in the body (for example, those involved in blood clotting and repair of damaged tissues), blood serum albumin); (e) manufacturing bile; (f) breaking down food and turning it into energy; (g) storing iron, vitamins, and other essential chemicals; (h) manufacturing, breaking down, and regulating numerous hormones, including sex hormones; (i) production of large quantities of purine nucleotides for bone marrow and neuronal tissue; (j) clearing the blood of particles and infections, including bacteria, etc. [18].

There are several metabolic states of the liver, the appearance of which depends on the type of food and the time passed after food consumption. The liver may convert one metabolite type into other metabolites to maintain metabolic homeostasis for different organs and tissues in the blood. For example, carbohydrates can be converted into fats and amino acids; fatty acids and amino acids can be converted into glucose. Of course, the metabolic properties of the liver also depend on the type of animal: herbivorous, carnivorous, or omnivorous (humans, rats, mice, and pigs). It should also be considered that everything that enters the blood from the stomach and gut goes directly into the liver through the portal vein. The liver is unique because the volume of venous blood (70%) exceeds that of the hepatic artery (30%). The hepatic vein tree then combines the blood flow from both inlet networks into a single stream. (see Figure 4). Thus, hepatocytes of the lobes, particularly those closer to the central vein, often experience hypoxia [19].

Liver and liver mitochondria can withstand up to 45 min of total ischemia if the animal was in the fed metabolic state and 30 min if the animal fasted 12 h before the experiment [20]. We have shown that a starved liver contains significantly higher levels of inorganic phosphate (Pi), which is a potent inhibitor of adenosine deaminase reaction, which, together with adenosine kinase, during ischemia participates in the production of ATP for the expense of degradation of adenine nucleotides [20]. We can distinguish several of the liver’s metabolic periods or states, depending on the time passed after the food consumption: (i) the absorptive period, (ii) the postabsorptive or fed state (i + ii = postprandial period), and (iii) the starved metabolic state, which has different degrees of metabolic changes depending on the longevity of fasting.

### 5.1. The Absorption Period

During absorption, the metabolites (carbohydrates, fatty acids, amino acids) released by the digestive organs are delivered through the portal vein to the liver cells. Long-chain fatty acids are packed into chylomicrons, enter lymphatic capillaries, and then transfer to the blood at the subclavian vein [21].

Fatty acids become activated to acyl-CoAs, carbohydrates are phosphorylated, and amino acids become converted into proteins or other metabolites. In rodents (rats and mice), the liver mitochondria can transport and activate short- and medium-chain fatty acids inside mitochondria without the participation of carnitine. Activation of fatty acids results in adenosine monophosphate formation: Fatty acyl + ATP + CoA → Fatty-acyl-CoA + AMP. Thus, AMP can accumulate in both the cytosol and the mitochondrial matrix. Therefore, if the liver mitochondria were isolated within 1–2 h post-feeding of the animal, the rate of oxidative phosphorylation would be meager because of the low content in the matrix of exchangeable mitochondrial adenine nucleotides ([ADP] + [ATP]) [22]. We have shown that isolated liver mitochondria contain a large amount of AMP during absorption. The AMP content in the mitochondrial matrix strongly depends on the type of diet. If the food is rich in fats, the level of AMP will be very high. Because of the potent inhibition of the adenine nucleotide translocase (ANT), oxidative phosphorylation will be inhibited, and the hepatocyte’s [ATP]/[ADP] and the phosphate potential ([ATP]/[ADP] + [Pi]) ratios also will be low. Liver mitochondria from the fed rats at the absorption period showed remarkable inhibition of succinate oxidation both in state 4 (resting respiration) and state 3 (phosphorylating respiration). Both carnitine and BSA did not affect the state 3 respiration of liver mitochondria from rats at the absorption period. This is because the limitation of the state 3 respiration is associated with the low exchangeable pool of adenine nucleotide in the matrix ([ADP] + [ATP]) due to the high concentration of AMP ([AMP]). In the in vitro experiments, the rate of oxidative phosphorylation can be restored by incubating mitochondria with α-ketoglutarate. During the conversion of α-ketoglutarate to succinate in the tricarboxylic acids (TCA) cycle, GTP is produced, which transphosphorylates ADP to ATP: GTP + ADP → GDP + ATP; then, the adenylate kinase reaction restores the exchangeable pool of adenine nucleotides in the transphosphorylation reaction: ATP + AMP → 2ADP [22].

### 5.2. The Postabsorptive or Fed Metabolic State

2–3 h after feeding, the isolated liver mitochondria show the highest rates of oxidative phosphorylation, low State 4 respiration, and, therefore, the highest respiratory control ratios (RCR). This is the so-called “fed” metabolic state of the liver. In this metabolic state, the liver has a high content of carbohydrates. Therefore, the pyruvate dehydrogenase complex is not inhibited, and the level of long-chain acyl-CoAs esters is low [22]. In the “fed” metabolic state, the liver mitochondria readily oxidize pyruvate + malate, citrate, isocitrate, and α-ketoglutarate, as well as glutamate or succinate.

The duration of the “fed” metabolic state is distinct for different species of rats [23] and differs between rats and mice. The smaller the animal, the shorter the fed metabolic state. For different strains of rats, the duration of the fed state may vary from 4 to 12 h from the time of the last feeding [23]. The time of transition from the fed to the starved metabolic state was characteristic for each strain of rats. In mice, the transition is much faster than in rats because they are much smaller animals (Kleiber’s Law). According to Kleiber’s Law, the metabolic rate for all organisms follows exactly the 3/4 power law of the body mass. It holds well from the smallest bacterium to the largest animal. The relation remains valid even down to the individual components of a single cell, such as the mitochondrion and the respiratory complexes [24,25]. Regarding humans, differences in the time of transition from the fed to the starved state could serve as a criterion for identifying whether people are prone to obesity and use this information to correct diet in terms of the optimal times of food consumption.

### 5.3. The Starved Metabolic State

The fed state is followed by the “starved” metabolic state when the liver switches from oxidation of carbohydrates to predominant oxidation of fatty acids and gluconeogenesis. In this metabolic state, hepatocytes accumulate long-chain acyl-CoAs, which inhibit oxidative phosphorylation and slightly uncouple liver mitochondria due to inhibition of the mitochondrial ANT and fixation of ANT in the “c” conformational state, which increases the conductivity of the inner membrane for H^+^ and K^+^ [22,26]. Significant variations in mitochondrial activity are due to variations in the duration of the absorption and fed metabolic states, which also depend on the type of diet and the species’ metabolic phenotype. Therefore, most researchers isolate liver mitochondria from animals starved overnight.

In the “starved” metabolic state, the mitochondrial pyruvate dehydrogenase complex is inhibited; therefore, the TCA cycle does not function normally, and the liver mitochondria do not oxidize pyruvate but metabolize acetyl-CoA from the fatty acid β-oxidation. However, in the in vitro experiments, if fatty acids are absent, liver mitochondria from the starved animals poorly oxidize citrate and other TCA-cycle intermediates. The only remaining substrates for respiration are glutamate + malate, succinate, and fatty acids. However, with the liver mitochondria, palmitoyl carnitine is oxidized slowly. Experiments with the brain, heart, and kidney mitochondria show that the high rate of palmitoyl-carnitine β-oxidation requires the presence of succinate, glutamate, or pyruvate. We must admit that we did not study the effects of supporting substrates on fatty acid oxidation by the liver mitochondria.

Using the overnight starved animals and isolated liver mitochondria oxidizing succinate, the rate of state 3 respiration significantly increased after adding carnitine [23]. This is because carnitine removed long-chain acyl-CoAs, powerful ADP/ATP carrier (ANT) inhibitors, in the cytosol. The increase of the state 4 respiration in the presence of carnitine may be explained by the fact that the transport of acyl-carnitines is an energy-dependent process, but mainly because acyl-CoA dehydrogenase feeds electrons to the membrane’s pool of CoQ, precisely as does Complex II during oxidation of succinate. As a result, the energy-dependent reverse electron transport and superoxide production are stimulated, increasing the state 4 respiration [9,26].

It should be remembered that liver mitochondria contain fewer respiratory complexes than heart or brain mitochondria. In addition, the high rate of ATP synthesis is not a primary function of the liver mitochondria. In hepatocytes, mitochondria are predominantly located close to the nucleus, reflecting the liver’s high synthetic activities. For the anabolic synthetic reactions and metabolism of xenobiotics, which occur in hepatocytes, the energy value of ATP or high [NADPH_2_] is much more critical than the ATP and NAD(P)H production rates. The energy states of the adenine nucleotide system have crucial regulatory functions in the metabolic processes in the liver. The values of the [ATP]/[ADP] ratio and [ATP]/[ADP] + [Pi]—phosphate potential reflect oxidative phosphorylation; the Energy Level, or the Energy Potential: [ATP] + 0.5[ADP]/[ATP] + [ADP]+ [AMP], which depends on both oxidative phosphorylation and adenylate kinase reaction to determine the viability of hepatocytes during ischemia or other harsh conditions [27].

### 5.4. Metabolic Liver Ischemia

In hepatocytes, several essential ATP-consuming reactions in the anabolic or anaplerotic (gluconeogenesis, synthesis of purines) and catabolic (urea synthesis) metabolic pathways occur in the mitochondrial matrix. Finally, almost all ATP produced by the liver mitochondria is utilized for biochemical reactions and transport of metabolites. This makes the liver functionally different from other organs, making it more challenging to investigate liver mitochondria. It should also be remembered that the metabolic features of liver cells and liver mitochondria are subject to significant species and strain variations within the same species.

Because the liver receives blood from the digestive system, the liver is particularly vulnerable to the toxic effects of some food metabolites and toxins. This situation most often occurs when a person consumes an alcoholic liquor containing up to 40% glycerol, a large amount of honey, or mead, an alcoholic beverage made of honey that contains a large amount of fructose. In the liver, both glycerol and fructose quickly undergo phosphorylation. However, both glycerol-3-phosphate and fructose-6-phosphate relatively slowly undergo further metabolism. Therefore, accumulation in hepatocytes of these phosphorylated metabolites exhausts the cell of inorganic phosphate (Pi) and thus diminishes the ATP content, similar to the effect of ischemia. This makes the liver vulnerable to damage caused by alcohol or xenobiotics. This is one of the possible adverse effects on the liver [28] and adrenal glands [29].

## 6. Digestive Efficiency

Human food contains three macronutrients that require digestion before they can be absorbed: fats, carbohydrates, and proteins. Digestion of food in the human digestive system combines different physicochemical processes, including food intake followed by physical and biochemical disintegration, intestinal enzymatic and bacterial digestion, membrane digestion, absorption, and transportation to the liver [30,31]. The human digestive system consists of the digestive tract and the accessory organs controlled by the neural network and the hormones [32]. Digestive juices are secreted by the salivary glands, gastric glands, pancreas, and liver with its adjuncts (the gallbladder and bile ducts). On average, humans can produce 1.5 L of saliva, 2 L of stomach secretions, and 0.5 L of bile solution in a day [31].

Humans spend between six and seven percent of their energy on food processing, compared with an average of 13% to 16% among other mammals. Preparing food with heat dramatically modified human anatomy and physiology, making them different from other mammal animals. Chimpanzees chew their food for 4 to 6 hours daily, while humans spend barely an hour doing so. Humans have smaller oral cavities, jaws, teeth, and weaker chewing muscles than anthropoid primates. Even our digestive system is proportionally shorter. However, because humans eat cooked food, the extraction of nutrients is more effective. Fats are absorbed with about 97% efficiency (e.g., if you eat 100 g of fat, you’ll absorb 97 g of it), animal-source proteins are about 90–95%, vegetable-source proteins can be in the 80% range, and carbohydrates vary drastically depending on their form, fiber content, etc. [33].

Evolving over hundreds of thousands of years, with primitive tools for obtaining food, humans rarely had an abundance of food and often experienced long periods of famine. Being hungry was a norm, whereas hunting required much physical and mental effort and energy. If the hunt was successful, people ate their fill and restored their carbohydrates and fat reserves. These transitions from hunger to abundant satiety require functional metabolic flexibility and economical consumption of internal resources. Thus, we can distinguish two primary metabolic states: well-fed rest and stressful hunting activity, accompanied by significant physical and mental performances. So far, we have yet to learn much about how different organs metabolically interact with one another under various physiological conditions. Therefore, we will look at the vast available data regarding metabolism from this point of view.

## 7. Caloric and Metabolic Efficiencies of the Three Macronutrients

Table 1 shows caloric values, storage amounts, and time of whole storage depletion for the three macronutrients. The human body uses these nutrients to build up tissues and as energy sources. Here, we will only briefly discuss the properties of each nutrient and then consider their metabolism separately in more detail. The table shows that only fatty acids exist in the human body in large quantities and can last for days. Structurally and functionally, fatty acids are divided into three groups.

### 7.1. Structural and Functional Diversity of Fatty Acids

#### 7.1.1. Long-Chain Fatty Acids (LCFAs)

LCFAs are the most abundant nutrients, with an unbranched chain of an even number of carbon atoms, from 14 to 28, that can be saturated or unsaturated. Most LCFAs are triglycerides that constitute fat tissue and phospholipids. Polyunsaturated fatty acids (PUFA) with three or more double bonds are usually not utilized as a source of energy. Most PUFAs exist in phospholipids in the C2 position that construct cell membranes and the inner mitochondrial membrane and also serve after lipolysis as substrates for specific enzymes to produce signaling molecules, such as prostaglandins [34]. In addition, PUFA is the primary target for oxidative damage by the protonated form of superoxide radical, perhydroxyl radical (HO_2_^•^), representing one of the most critical mechanisms of aging [35].

#### 7.1.2. Medium-Chain Fatty Acids (MCFAs)

MCFAs are saturated fatty acids that present six to 12 carbon atoms. Caproic acid (C6:0), caprylic acid (C8:0), capric acid (C10:0), and lauric acid (C12:0) are examples of MCFAs. They are triglycerides in milk, dairy products, coconut oil, and palm kernel oil [36,37]. Those triglycerides are often completely hydrolyzed to yield free fatty acids by lipases in the gastrointestinal tract. When absorbed directly, medium-chain triglycerides (MCTs) enter the blood circulation and are carried to the liver to oxidize to ketones [38]. Compared with the triglycerides containing long-chain fatty acids, MCTs have a lower melting point, a smaller molecular size, provide slightly lower energy (8.4 vs. 9.2 kcal/g), and are liquid at room temperature (reviewed in [38]). MCFAs are also crucial as signaling molecules [35]. Dietary MCFAs are not the primary source of our body MCFAs but are mainly produced from β-oxidation of LCFA in the target organ cell and tissue [39]. Interestingly, in the Southern Community Cohort Study, it was established that in a predominantly low-income United States population, an increased milk dietary C12:0 intake was associated with a substantially reduced risk of colorectal cancer only among White individuals but not in Black individuals [40].

#### 7.1.3. Short-Chain Fatty Acids

Short-chain fatty acids (SCFAs) are carbon chains of one to six atoms produced in the gut through microbial fermentation of plant polysaccharides and dietary fibers. They are a subset of saturated fatty acids, including acetate, propionate, butyrate, pentanoic (valeric), and hexanoic (caproic) acids. Some researchers classify hexanoic (caproic) acids as short-chain [41] and some as medium-chain fatty acids [36].

SCFAs are a subset of fatty acids the gut microbiota produces during the fermentation of partially and nondigestible polysaccharides. The highest levels of SCFAs are found in the proximal colon, where they are predominantly used locally by enterocytes or transported across the gut epithelium into the bloodstream. Two central SCFA signaling mechanisms have been identified: inhibition of histone deacetylases (HDACs) and activation of G-protein-coupled receptors (GPCRs) [41]. Acetate, propionate, acetoacetate, and butyrate are the most common SCFAs [42]. During lipid digestion, SCFAs and medium-chain fatty acids are primarily absorbed through the portal vein [43].

#### 7.1.4. Not Common Low-Abundance Fatty Acids

**Odd-chain fatty acids** undergo β-oxidation in the same metabolic pathways as even-chain fatty acids. However, the final spiral of β-oxidation yields one acetyl-CoA molecule and one propionyl CoA molecule. This three-carbon molecule undergoes enzymatic conversion to succinyl CoA, thus bridging the tricarboxylic cycle and fatty acid oxidation. However, fatty acids with odd carbons aliphatic chains are rare in Nature [44].

**Very-long-chain fatty acids (VLCFA) beta-oxidation in peroxisomes** occur similarly to mitochondrial β-oxidation. However, some key differences exist: (1) different genes encode fatty acid oxidation enzymes in peroxisomes; (2) in the first step of the peroxisomal pathway, the enzyme responsible for the production of a double bond between the alpha and beta carbon is an oxidase that donates electrons to molecular oxygen producing hydrogen peroxide. The remaining three steps are similar to the mitochondrial β-oxidation steps [45]. Carnitine may also transfer short- to medium-chain fatty acids from peroxisomes to the mitochondrial matrix to complete oxidation [46].

**Branched-chain fatty acids require additional** enzymatic modification to enter the α-oxidation pathway within peroxisomes. Phytanic acid cannot be synthesized in the human body; it is solely derived from exogenous dietary sources as a byproduct of chlorophyll degradation. While chlorophyll in vegetables is a potential source of phytanic acid, it cannot be digested by humans. In contrast, ruminant animals, with the help of their gastric flora, can absorb the chlorophyll-bound phytol and metabolize it to phytanic acid. The primary sources of phytanic acid are milk products and meat of ruminant animals, such as beef, lamb, and veal, as well as predatory fish (e.g., cod and tuna) [47]. The daily intake with a regular diet is 50–100 mg. Because phytanic acid is not metabolized in patients with Refsum’s disease and the only source of phytanic acid in humans is dietary, restriction of oral intake of phytanic acid and, to a lesser extent, of phytol, which can be converted into phytanic acid, has been advised and found beneficial. Specifically, dietary intake of dairy products, fats, and meats from ruminant animals, all of which contain phytanic acid, must be markedly curtailed. Caution should be observed to avoid starvation diets, as they can cause rapid mobilization of body stores of phytanic acid with a marked increase in the elevation of serum phytanic levels, acute toxicity, cardiac arrhythmias, and possible cardiac arrest. Phytanic acid, 3,7,11,14-tetramethyl hexadecanoic acid, requires additional peroxisomal enzymes to undergo β-oxidation. Phytanic acid initially activates phytanyl CoA; then, phytanyl CoA hydroxylase (α-hydroxylase), encoded by the *PHYH* gene, introduces a hydroxyl group to the alpha carbon [48]. The alpha carbon-hydroxyl bond then undergoes two successive rounds of oxidation to pristanic acid. Pristanic acid undergoes β-oxidation, which produces acetyl CoA and propionyl CoA in alternative rounds. As with peroxisomal beta-oxidation of very long-chain fatty acids (VLCFAs), this process generally ends when the carbon chain length reaches six to eight carbons. The molecule is shuttled to the mitochondria by carnitine for complete carbon dioxide and water oxidation [48].

**Omega-oxidation of fatty acids** in the endoplasmic reticulum primarily functions to hydroxylate and oxidize fatty acids to dicarboxylic acids to increase water solubility for excretion in the urine. This enzymatic conversion relies on the cytochrome P450 superfamily catalyzing reactions between xenobiotic compounds and molecular oxygen [49]. Deficiencies in some enzymes of ω-fatty acid oxidation may result in their accumulation. Thus, upregulation of ω-oxidation and increased serum and urine medium-chain dicarboxylic acids can diagnose certain deficiencies.

### 7.2. Amino Acids

Amino acid reserves are practically absent and are formed continuously due to the digestion of food proteins, anaplerotic reactions in mitochondria, and degradation of proteins from the body’s dying cells.

### 7.3. Carbohydrates

Unlike fatty acids, glucose storage is minimal, and glucose is in great demand for the biosynthesis of many other vital molecules: nucleic acids, purine, pyrimidine nucleotides, glycolipids, glycoproteins, etc. Therefore, glucose usage as an energy source is always limited. However, under conditions of highly increased gluconeogenesis, for example, during prolonged high physical work, aerobic glycolysis may be a substantial source of ATP [50]. Kidneys practically do not utilize glucose as a source of energy. The absolute demands for glucose have only erythrocytes that consume at least 240–360 g of glucose in a day by a person of average size. Increased gluconeogenesis is one of the reasons that the blood glucose level remains constant even during long-term starvation. The average fasting blood glucose concentration (no meal within the last 3 to 4 h) is between 80 and 90 mg/dL. On average, postprandial blood glucose may rise to 120–140 mg/dL, but the body’s feedback mechanism returns the glucose to normal within 2 h. During starvation, the liver and kidneys provide glucose to the body through gluconeogenesis, synthesizing glucose from lactate and amino acids [51,52]. Even the central nervous system satisfies its energy demands by more than 20% for the expense of β-oxidation of the LCFAs [11,53]. It must be remembered that a significant portion of the metabolites derived from glucose and amino acids are constantly consumed as supporting substrates during fatty acids β-oxidation. Since the carbohydrate stores are small, they must be constantly replenished by gluconeogenesis in the liver and kidneys. During prolonged starvation or diabetes, the supporting β-oxidation substrates become scarce. In the liver, acetyl-CoA excess converts into ketone bodies, serving as alternative substrates for mitochondria in the heart, brain, and skeletal muscles. The old myth is that the brain only consumes glucose to support energy needs. However, most of the lactate and neuromediators glutamate and γ-aminobutyric acid, which are also used by synaptic mitochondria as energy substrates, are synthesized by the astrocytes from the carbon atoms of fatty acids and for the expense of energy derived during β-oxidation of fatty acids. Synaptic mitochondria also gladly oxidize fatty acids in the presence of supporting substrates [11]. Type 2 diabetes is characterized by insulin resistance and chronic hyperglycemia. The kidneys ameliorate hyperglycemia by converting part of the glucose excess into lactate in the proximal tubules, which is consumed as the energy source by neurons and heart muscles. These are excellent examples of the metabolic interactions between organs.

## 8. A Deeper Understanding of the Body’s Metabolism Requires Fundamental Paradigm Changes

### 8.1. Description of Old Paradigms

Georg Franz Knoop discovered fatty acid β-oxidation in 1904 [54]. Since then, many details of β-oxidation have been found. According to the generally accepted concepts today, mitochondrial β-oxidation of fatty acids occurs in four stages, preceded by several steps of fatty acid release from triglycerides, transport, activation, etc., described in detail in textbooks and reviews [46,54,55]. Figure 5 schematically illustrates these events.

Fatty acids utilized for cardiac fatty acid β-oxidation primarily originate from either plasma fatty acids bound to albumin or from fatty acids contained within fat tissue surrounding the heart or very-low-density lipoproteins (VLDL) triacylglycerol (TAG). Fatty acid β-oxidation is the process by which fatty acids are broken down to produce energy. Fatty acids are taken up by the heart via diffusion or CD36/FATP transporters. Once inside the cytosolic compartment of the cardiac myocyte, fatty acids (bound to fatty acid-binding proteins) are esterified to fatty acyl CoA by fatty acyl Coenzyme A synthase (FACS). Then, carnitine palmitoyltransferase 1 (CPT1) converts the long-chain acyl-CoA to long-chain acylcarnitine. Carnitine acyltransferase (CAT) transports the acylcarnitine across the inner mitochondrial membrane. CPT2 then converts the long-chain acylcarnitine back to long-chain acyl-CoA inside mitochondria, and the long-chain acyl-CoA then enters the fatty acid β-oxidation pathway [46].

The actual cyclic process of β-oxidation of long-chain fatty acids is believed to occur in four steps in the mitochondrial matrix, producing three energy storage molecules per round of oxidation, namely, one NADH, one FADH_2_, and one acetyl CoA molecule. Figure 6 shows the sequence of the enzymatic reactions of the fatty acids β-oxidation spiral following Old Paradigms [46].

Step 1. The first enzyme, acyl CoA dehydrogenase, is specific to chain length, like other enzymes involved in handling fatty acids. Members of this enzyme family include long-chain, medium-chain, and short-chain acyl CoA dehydrogenases (LCAD), (MCAD), and (SCAD), respectively. These enzymes catalyze the formation of a trans double bond between the alpha and beta carbons on acyl CoA molecules by removing two electrons to produce one molecule of FADH_2_.

Step 2. The enzyme enoyl CoA hydratase performs a hydration step of the double bond between the alpha and beta carbons, adding a hydroxyl (OH^−^) group to the beta carbon and a proton (H^+^) to the alpha carbon.

Step 3. Following hydration, the enzyme beta-hydroxyl acyl CoA dehydrogenase removes two electrons and two protons from the hydroxyl group and the attached beta carbon to oxidize the beta carbon and produce a molecule of NADH.

Step 4. The final step in β-oxidation involves CoA cleavage of the alpha and beta carbon bond. This step is catalyzed by beta-keto thiolase. The reaction produces one molecule of acetyl CoA and a fatty acyl CoA that is two carbons shorter. The process may repeat until the even-chain fatty acid has completely converted into acetyl CoA.

Under the new paradigms, all enzymes participating in the β-oxidation of FAs are mitochondrial, located in the inner mitochondrial membrane, and assembled into two polyenzymatic complexes. The first step of β-oxidation, catalyzed by acyl CoA dehydrogenase, is a complex of three FAD-containing enzymes that can be disassembled into individual enzymes (see Figure 7) [56]. There are three isoenzymes of acyl-CoA dehydrogenase, each for a specific length of the aliphatic chain: one for long-chain FAs (12–20C), the other for the middle-chain FAs (6–12C), and the third one for the short-chain FAs (4–6C). The overall structure is similar across different chain-lengths specific acyl-CoA dehydrogenases. Complete oxidation of long-chain fatty acids requires the sequential work of the three isoforms of acyl-CoA dehydrogenases [57]. The reduction of the membrane CoQ to CoQH_2_ occurs very quickly, considering that acyl-CoA dehydrogenases form homodimers that further associate into tetramers, with each monomer containing a single FAD molecule [58].

The following three steps of β-oxidation, namely, 2-enoyl coenzyme A (CoA) hydratase, long-chain 3-hydroxy acyl-coenzyme A dehydrogenase, and long-chain 3-ketoacyl CoA thiolase, are assembled into a durable complex called a trifunctional protein [59,60]. The biological unit of the protein is α2β2. The two beta-subunits make a tightly bound homodimer at the center, and two alpha-subunits are bound to each side of the β2 dimer, creating an arc, which binds on its concave side to the mitochondrial inner membrane. The catalytic residues in all three active sites are arranged similarly to those of the corresponding soluble monofunctional enzymes [61]. Therefore, from a functional point of view, the cyclic β-oxidation occurs very quickly in two steps only. We can add that the two enzyme complexes (Figure 7) of β-oxidation are physically associated with the mitochondrial respirasome comprised of three supercomplexes [62].

In contrast to the physiological studies, the isolated mitochondria from practically all organs do not oxidize fatty acylcarnitines at a high rate. For this reason, β-oxidation of LCFAs and MCFAs remains largely unstudied at the mitochondrial level. Nevertheless, several important discoveries have been made in decades, allowing us to suggest a new hypothesis that resolves the above-mentioned problems [12].

### 8.2. Superstructural Functional Organization of Fatty Acid Metabolism

Brand et al. have shown that in the organs where the β-oxidation of long-chain fatty acids is the primary energy source, the highest stationary levels of ubiquinol are maintained [63,64]. Moreover, at a high level of ubiquinol, succinate dehydrogenase reverses the flow of electrons from ubiquinol into mitochondria and reduces fumarate to succinate [63,64]. We have shown that with the isolated mitochondria from the heart, kidney, and brain (Figure 3), the rate of palmitoyl-carnitine oxidation in state 3 increased several-fold in the presence of mM concentration of pyruvate, glutamate, or succinate [6,8]. We have suggested that the actual stimulator of reverse electron transport is succinate, which can be formed from glutamate or pyruvate in the transamination reactions [8,12]. We suggest that the supporting substrates work as allosteric activators of the reverse electron transport through complex II.

One cycle of β-oxidation produces one molecule of acetyl-CoA, one NADH, and one molecule of CoQH_2_. The highly hydrophobic molecule of ubiquinol instantly becomes oxidized by the minor supercomplex of the respirasome (see Figure 8). In contrast, the mitochondrial energy-dependent transhydrogenase transfers the hydrogen of the matrix NADH to the cytosolic NADP^+^, reducing it to NADPH. By reversing the electron transfer via SDH (complex II) from QH_2_ to the matrix, β-oxidation involves acetyl-CoA and the tricarboxylic acid cycle into synthetic and anaplerotic reactions. Namely, gluconeogenesis in the liver and kidneys, aerobic glycolysis with lactate formation in skeletal muscles and brain astroglia, and other energy-consuming processes necessary for maintaining an organ’s cells.

Acetyl-CoA is a carrier of two-carbon units. It is a critical metabolite due to its intersection with many metabolic pathways and transformations. Through the changes in protein acetylation, cells monitor acetyl-CoA levels as a crucial indicator of their metabolic state. High nucleocytosolic acetyl-CoA amounts are a signature of a “growth” or “fed” state and promote the utilization of acetyl-CoA for lipid synthesis and histone acetylation. In contrast, under “survival” or “fasted” states, acetyl-CoA is preferentially directed into the mitochondria to mitochondrial catabolic activities promoting the production of ATP and ketone bodies [65]. The fate of acetyl-CoA is organ-specific and depends on mitochondrial respiratory activity and acetylation/deacetylation of key enzymes. For example, in the fasted liver, high levels of Ac-CoA due to high activation of fatty acids and a relatively low rate of β-oxidation result in the stimulation of lipid biosynthesis, ketone body production, and the diversion of pyruvate metabolism toward gluconeogenesis and away from oxidation. Low levels of acetyl-CoA exert opposite effects [66,67]. Acetylation changes the properties of molecules. Acetyl-CoA is necessary to synthesize acetylcholine, acetyl glutamate, acetyl aspartate, and N-acetyl amino sugars and eliminate some xenobiotics [66].

### 8.3. The Necessity of Substrates Alternative to Fatty Acids as Energy Sources

The oxygen-consumption rate during active oxidative phosphorylation (metabolic state 3) reflects the rate of ATP production by mitochondria, which in experiments with isolated mitochondria strongly depends on the substrate or substrate mixtures provided to mitochondria. The observed maximal respiration rates in state 3 for mitochondria isolated from the rat heart, brain, and kidney were 230–240 nmol O_2_/min./mg mitochondrial protein when the mitochondria oxidized palmitoyl-carnitine plus pyruvate, glutamate, or succinate as supporting substrates [6,7,8,11]. Here, we have to note that the functional loads for the brain, particularly for the skeletal muscle and heart mitochondria, may change in a wide range (see Figure 9). In contrast, kidney mitochondria work at a relatively constant pace. Kidney mitochondria do not have the intrinsic inhibition of SDH. With 5 mM succinate, the respiration rate in state 3 was at maximum and equal to palmitoyl-carnitine + succinate [8]. Succinate is also the most effective in stimulating the oxidative phosphorylation rate with palmitoyl-carnitine and octanoyl-carnitine. In the absence of carnitine, the kidney mitochondria poorly oxidize both palmitoyl and octanoyl. With octanoyl-carnitine + succinate, the state 3 respiration of mouse kidney mitochondria was significantly higher than with palmitoyl-carnitine: 400 versus 280–300 nmol O_2_/min/mg/mitochondrial protein [8].

From the above data, we can deduce that the rate of ATP production, supported by the β-oxidation of the long-chain fatty acids, has its limits. We suggest that the ATP production rate (oxidative phosphorylation) and/or reverse electron transport on SDH limit the oxygen consumption rate, not the β-oxidation itself. Therefore, when the heart increases ATP consumption almost tenfold during intensive training, this cannot be achieved by a tenfold increase in fatty acid β-oxidation. Conceivably, some alternative oxidative mechanisms for ATP production must also be activated. Researchers have long considered glucose as an alternative to the fatty acid substrate or an additional substrate [69,70,71].

### 8.4. Metabolic Flexibility

Energy metabolism and substrate consumption for ATP production change depending on environmental conditions, such as temperature, physical activity, or metabolic state, such as feeding vs. fasting. This adaptive switching from one substrate to another was designated as “metabolic flexibility” [72,73,74,75]. There are many definitions of flexibility: “the ability of an organism to respond or adapt according to changes in metabolic or energy demand as well as the prevailing conditions or activity” [76]. Smith et al. [73] defined metabolic flexibility “as an adaptive response of an organism’s metabolism to maintain energy homeostasis by matching fuel availability and demand for periodic fasting, varying meal composition, physical activity, and environmental fluctuations”. Other researchers defined metabolic flexibility as “the capacity for the organism to adapt fuel oxidation to fuel availability so ATP synthesis can match its demand” [74,75]. And even metabolic changes with aging [77].

Our discussions will use data on the adaptive metabolic changes in skeletal muscle mitochondria during moderate and intensive exercise. Skeletal muscles display the highest range of functional activity (Figure 9), and the accompanying metabolic changes have been studied the most. During endurance exercise, the contribution of LCFA to energy metabolism increases with time, whereas when the intensity of exercise increases, carbohydrates cover the energy need more and more [50]. In addition to glucose, Ahlborg et al. also examined the dynamics of changes in fatty acids and other metabolites. The authors have provided important information about changes with time in substrate metabolism during prolonged moderate physical exercise [78]. The authors studied the arterial concentrations and substrate exchange across the leg and splanchnic vascular beds for glucose, lactate, pyruvate, glycerol, individual acidic and neutral amino acids, and free ^14^C-labeled oleic acid during 4 hours of exercise at approximately 30% of maximal oxygen uptake. The arterial glucose concentration was constant for the first 40 min of exercise but fell progressively to levels 30% below basal. The arterial insulin level decreased continuously, while the arterial glucagon concentration rose fivefold after 4 hours of exercise. The uptake of glucose and oleic acid by the legs was markedly increased during exercise. As the exercise was continued beyond 40 min, the relative contribution of oleic acid to total oxygen metabolism rose progressively to 62%. In contrast, the contribution from glucose fell from 40% to 30% between 90 and 240 min. The leg output of alanine increased as the exercise progressed. Splanchnic glucose production rose 100% above the basal levels. Glucose uptake by the legs increased for the first 40 min but failed to keep pace with peripheral glucose utilization after that. Splanchnic uptake of gluconeogenic precursors (lactate, pyruvate, glycerol, alanine) had increased 2- to 10-fold after 4 hours of exercise. The authors concluded that during prolonged exercise at a low work intensity, (a) blood glucose levels fall because hepatic glucose output fails to keep up with increased glucose utilization by the exercising legs; (b) a large portion of hepatic glycogen stores is mobilized, and the increased splanchnic glucose output was derived from gluconeogenesis; (c) blood-borne substrates in the form of glucose and oleic acid account for a significant part of leg muscle metabolism. The contribution from FFA was increasing progressively, and (d) augmented glucagon secretion may play an essential role in the metabolic adaptation to prolonged exercise by its stimulatory influence on hepatic glycogenolysis and gluconeogenesis [78].

Thus, during prolonged physical exercise, the increased utilization of fatty acids is accompanied by increased glucose consumption and gluconeogenesis. Without information about the actual properties of β-oxidation of long-chain and middle-chain fatty acids in mitochondria, researchers concentrated on the investigations of ketone bodies and short-chain fatty acids, particularly on glucose as energy substrates [3,37,41,42,69]. Under normal conditions, short-chain fatty acids contribute little to the overall energy metabolism during exercise because they are produced by the intestine microbiota, where they are mostly consumed and fulfill their regulatory functions [41,79,80].

## 9. Aerobic Glycolysis Products Lactate and Pyruvate Are Crucial Alternative Energy Sources

Many reviews provide an instructive retrospective history of studies on the role of glucose in energy metabolism. We cite only a few that discuss various aspects of energy metabolism [11,69,71,81,82,83,84,85,86]. We list some of the main conclusions of these studies.

(1)Lactate and pyruvate cross the cell membranes via a monocarboxylate transporter (MCT) with well-defined properties but an undefined molecular structure [87].(2)The working muscle produces lactate, which is a crucial fuel energy source for the muscle; lactate is also released from the working muscle and fuels other organs such as the heart [88,89] and brain [90], and serves as a gluconeogenic substrate to the liver and kidney [52,91].(3)In a working skeletal muscle, glucose and lactate are continuously produced under aerobic conditions; lactate is also extracted from the blood [92,93].(4)In resting and working human skeletal muscles and the heart, aerobic glycolysis simultaneously produces lactate and undergoes oxidative disposal [94]. Similar events were observed also in the brain [90].(5)Lactate is the most essential gluconeogenic precursor during rest and exercise [52,91].(6)Skeletal muscles, brain, and heart mitochondria prefer lactate over pyruvate to produce ATP [71,95].

The story of lactate and its role in physiology and medicine may be a century old, but it has changed dramatically in the last 3 decades [83,86,95]. In contemporary physiology, lactate is seen as a significant metabolic intermediate with wide-ranging impacts on energy substrate utilization, cell signaling, and adaptation; it is the central metabolite for metabolic integration [83,85]. In resting and exercising human muscles, lactate concentration exceeds pyruvate 10 and 100 times, respectively. Therefore, muscle tissue, particularly during exercise, is a significant source of circulating lactate [96]. Like most keto-carboxylic acids, pyruvate undergoes non-enzymatic decarboxylation in the presence of H_2_O_2_ to form acetate, carbon dioxide, and water [97]. It was an important discovery that in vivo, the product of glycolysis is always lactate, not pyruvate [98], and that glucose and glycogen catabolism proceed to lactate production under fully aerobic conditions [98,99]. Traditionally, the end product of glycolysis has been viewed as dependent on tissue oxygenation: pyruvate in the presence of adequate oxygenation and lactate—in the absence of proper oxygenation. Rogatzki et al. [98] considered differences in the enzyme activities, namely that the lactate dehydrogenase (LDH) reaction is at near-equilibrium and has much higher activity than the regulatory enzymes of the glycolytic and oxidative pathways, as the main reason that lactate is always the end product of glycolysis. According to our hypothesis, presented here, there are two reasons why in vivo glycolysis ends up in the formation of lactate: (1) fatty acid β-oxidation generates a high NADH/NAD+ ratio, and (2) glycolytic pyruvate is unable to enter the TCA cycle. All these facts were impossible to understand and explain based on classical enzymology and thermodynamics paradigms.

In the irreversible thermodynamics (IT) paradigms developed by Ilya Prigogine [100], living organisms are defined as dissipative structures within thermodynamic systems far from equilibrium. From the paradigms of IT, the metabolic processes represent steady-state systems in which individual metabolic processes have a structural organization and one or more points of irreversibility. During the last few decades, data have accumulated indicating that all metabolic pathways are organized into polyenzymatic complexes (β-oxidation), supercomplexes (respirasome) [59,60], and even more complicated physiological structures comprising several polyenzymatic complexes and supercomplexes [12,62].

Brooks pointed out that during intensive muscle work, lactate is simultaneously produced by aerobic glycolysis and consumed as an energy source [85,95]. Both metabolic processes are spatially separated and irreversible. The Embden–Meyerhof–Parnas (EMP) pathway comprises 11 cytosolic enzymes interacting to metabolize glucose to lactic acid [101] that, in principle, do not depend on pO_2_ in the cytoplasm [99]. On the other hand, lactate is irreversibly consumed in mitochondria through oxidation or carboxylation. Roosterman [101] points out that lactate dehydrogenase (LDH) complexes irreversibly reduce pyruvate/H^+^ to lactate or irreversibly catalyze the opposite reaction, oxidation of lactate to pyruvate/H(^+^); both LDH complexes are spatially separated and driven by the main point of irreversibility represented by the minor supercomplex of respirasome oxidizing the inner membranes CoQH_2_ (Figure 10).

Figure 10 shows how fatty acids β-oxidation governs the glucose-lactate-glucose cycle and other anabolic and anaplerotic metabolic pathways. The one cycle of fatty acid β-oxidation reduces one mitochondrial CoQ to CoQH_2_, one NAD^+^ to NADH, and produces one molecule of acetyl-CoA. CoQH_2_ is instantly irreversibly oxidized by the minor supercomplex of respirasome, thus creating a high steady-state energization of mitochondria. The tetrameric structure of the two β-oxidation enzymatic complexes is specifically designed to rapidly reduce the inner membrane pool of CoQ. The physical association of the β-oxidation complexes with the supercomplexes of the respirasome creates and maintains a high CoQH_2_/CoQ ratio in the inner membrane, and this reverses the flow of electrons via the SDH (complex II) from CoQH_2_ to the TCA cycle [63,64]. Thus, during β-oxidation, the high level of reduction of the TCA cycle and the rapid production of acetyl-CoA prevent glycolytic pyruvate oxidation in the TCA cycle and promote gluconeogenesis by stimulating pyruvate carboxylation and other anaplerotic reactions by utilizing various cycle intermediates. Because β-oxidation generates acetyl-CoA with high metabolic pressure due to the highly irreversible oxidation of ubiquinol, the pyruvate, generated by aerobic glycolysis, becomes irreversibly reduced to lactate.

Oxidation of the inner membrane’s CoQH_2_ by the minor supercomplex of the respirasome is a highly irreversible reaction. This is the main point of irreversibility that makes irreversible the anabolic (fatty acids synthesis, gluconeogenesis) and anaplerotic (synthesis of glutamate). The second point of irreversibility of the TCA cycle arises from the high concentration of acetyl-CoA produced by the FA β-oxidation and metabolized in the TCA cycle. As a result, pyruvate, formed in aerobic glycolysis, cannot enter the TCA cycle and becomes reduced to lactate or undergoes carboxylation with the formation of oxaloacetate, which enhances gluconeogenesis. This metabolic pathway is also irreversible because of the constant supply of initial substrate oxaloacetate, the removal of the final product (glucose) in aerobic glycolysis, and high redox and energy potentials in mitochondria generated by the FAs β-oxidation.

Because in the skeletal muscles, gluconeogenesis increases with time and exercise intensity [78,96], we can suggest that aerobic glycolysis may serve as an additional source of ATP to support β-oxidation during increased energy demands. Although glycolysis produces only two ATP, many glycolytic enzymes can match fatty acid β-oxidation in ATP production, particularly in the white muscle fibers [102]. Due to the high rate of β-oxidation, lactate is produced in excess, enters the blood circulation, and serves in the liver and kidney as a substrate for gluconeogenesis or oxidized in the heart and brain mitochondria. The membrane-bound lactate dehydrogenase (LDH) catalyzes lactate oxidation to pyruvate and H^+^. The mitochondrial pyruvate carrier transports pyruvate into the mitochondrial matrix [103], whereas H^+^ is transported into the matrix via the malate-aspartate shuttle, similar to synaptic mitochondria [11].

Thus, it can be assumed that the primary purpose of β-oxidation is to create thermodynamic conditions for directing the work of the TCA cycle towards anabolism and maintaining the aerobic cycle glucose-lactate-glucose as an alternative source of ATP under conditions of increased energy needs for the body activity. Brooks compared the metabolic role of lactate with the rise of Phoenix [95]. We can add that fatty acids β-oxidation creates the flame from which the Phoenix rises.

## 10. The Role of Fatty Acid Oxidation Products in Cell Synthesis, Signaling, and Homeostasis

In addition to being a critical source of cellular energy production, fatty acids play an indispensable role in many cellular functions [104]. The structural function of fatty acids is very well-recognized, and it is typically considered part of anabolic metabolism, where fatty acids are synthesized and used as a “building block” of cellular membranes. Fatty acids undergo esterification into a neutral lipid to be incorporated into lipophilic membranes. Polyunsaturated fatty acids (PUFA) are part of the membrane’s phospholipids. PUFA are particularly important for the inner membrane phospholipids cardiolipin and phosphatidylethanolamine, which hold the polyenzymatic structures within the inner membrane [12,35]. It is important to remember that there are only two “essential” fatty acids, linoleic acid, an ω-6 fatty acid, and α-linolenic acid, a ω-3 fatty acid, which the human body cannot produce and can get only from food [105]. All other 500 types of fatty acids are synthesized in the human body, and this requires not only energy but also fatty acid metabolism, including mitochondrial fatty acid β-oxidation, which produces a critical building block for fatty acid synthesis, acetyl-CoA [106]. It is important to note that impaired mitochondrial fatty acid β-oxidation leads to the accumulation of free- and esterified fatty acids inside cells, which can alter cellular function and promote inflammation and disease progression. The third important role is synthetic function. Metabolism of fatty acids provides numerous critical building blocks for many cellular synthetic pathways. Most synthetic pathways require ATP and NADPH, which are the products of mitochondrial fatty acid β-oxidation. Furthermore, acetyl-CoA is a precursor for the biosynthesis of cholesterol, ketogenesis, amino acids, isoprenoids, and sugars [107]. Interestingly, medium-chain fatty acids (C8–C12), produced during long-chain fatty acid β-oxidation, are critical for cellular biosyntheses, such as lipoic acid [108]. Finally, fatty acids and their products are essential in cell signaling, influencing gene expression, cell motility, hormone secretion, and prostaglandins formation. Many fatty acids can be conjugated with the protein lysine residues (i.e., palmitoylation, octanoylation) and activate G protein-coupled receptors [109]. Medium-chain fatty acids can act at the plasma membrane, activating GPR40 and GPR84, or in the cytosol, activating nuclear receptor peroxisome proliferator-activated receptors and regulating the gene transcription in the nucleus [110]. Another product, acetyl-CoA, can directly influence acetylation and thus, directly and indirectly, affect the epigenetic regulation of gene expression [111]. These data unambiguously show that fatty acid β-oxidation is not just an ATP cow (although indispensable) but is also critically important to multiple cellular functions. Therefore, impaired fatty acid oxidation may contribute to cellular and organ dysfunction both in ATP-dependent and independent fashion.

## 11. Discussion and Final Remarks

The discrepancies between many old ideas about metabolism and many new facts and discoveries prompted us to rethink the interactions of fatty acids β-oxidation with other metabolic pathways of ATP production. For example, changes in metabolic processes include new ideas about aerobic glycolysis, previously considered the prerogative of malignant tumors [112,113], and the metabolic role of lactate [83,85,86]. The discovery of the respirasome [114,115] forced researchers to reconsider old facts and ideas about the respiratory chain and substrates for mitochondrial respiration. Despite recognizing that long-chain fatty acids are the basis of human energy metabolism, β-oxidation of fatty acids by end-organ mitochondria has remained largely unexplored. This was because palmitoyl-carnitine, a common long-chain substrate, is oxidized very slowly when used with malate only. Recently, it was discovered that in the presence of succinate, glutamate, or pyruvate, the rate of state 3 respiration increases several-fold [6,8]. Brand and colleagues [63,64] have discovered that during fatty acid β-oxidation, electrons are reversed from the membrane’s pool of CoQH_2_ to the mitochondrial TCA cycle, reducing fumarate to succinate.

The discoveries presented above have allowed us to formulate a hypothesis that may enable us to understand how different types of metabolism interact to perform specific functions in different organs in states of relative rest or intense stress. Metabolic flexibility describes the ability of individuals to switch between crucial fuel energy substrates in response to changing physiological conditions such as obesity and diabetes. In the case of lactate and lipolysis in adipose tissue during strenuous exercise, there is an inverse relationship between blood lactate and plasma-free fatty acid concentration [116]. The crossover concept replaced the hypothesis of switching the energy source from carbohydrates to lipids. The crossover point is the power output at which energy from carbohydrate fuels predominates over energy from lipids, with further increases in power eliciting a relative increment in carbohydrate utilization and a decrement in lipid oxidation [95,116]. It has been shown that lactate production and utilization may occur irreversibly and simultaneously in the same organ and even cells [95]. It was unclear, though, what the source of glucose was and the origin of the irreversibility.

Earlier, it has been shown that with the NAD-reducing substrates (i.e., pyruvate), the rate of mitochondrial respiration is limited by the activity of the NADH-dehydrogenase of the complex I. Therefore, the respiration and ROS-generation rates in mitochondria oxidizing NAD-dependent substrates are at a minimum and do not depend on the energy state of the mitochondria [117]. For this reason, the two large supercomplexes of the respirasome, containing complex I, cannot supply enough electrons to drive the cell’s energy-dependent functions. On the contrary, the minor supercomplex of the respirasome lacks complex I and directly oxidizes the inner membrane pool of CoQH_2_, which SDH generates during succinate oxidation or by fatty acid β-oxidation (Figure 8). Meanwhile, the function of the two large complexes is to transfer the energy of the mitochondrial NADH/NAD^+^ through the energy-dependent transhydrogenase to the cytosolic NADPH/NADP+ system [12].

The minor supercomplex of the respirasome comprises a dimer of complex III and two dimers of complex IV [114,115]. Therefore, energy is dramatically released as heat during respiration, and the local temperature may be near 50 °C [118]. This central point of irreversibility drives most other mitochondrial irreversible metabolic pathways during fatty acid β-oxidation.

## 12. Conclusions

In living cells, there are only two major sources of ATP: glycolysis, which is independent of oxygen, and mitochondrial oxidative phosphorylation. However, neither of these ATP-production pathways can satisfy the maximum demand for ATP. Existing data support the hypothesis that fatty acid β-oxidation creates conditions for a significant acceleration of glycolytic ATP production. We suggest that fatty acid β-oxidation significantly accelerates the cycle: glucose-aerobic glycolysis-lactate-gluconeogenesis-glucose.

## Figures and Tables

**Figure 1 ijms-25-12740-f001:**
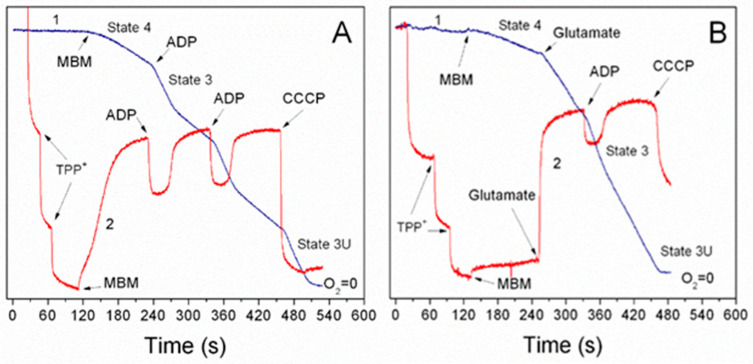
Oxygen consumption and membrane potential by mouse (strain FVB) brain mitochondria (MBM) isolated with and without BSA. Line 1—O_2_ consumption; Line 2—membrane potential. (**A**). Forebrain mitochondria were isolated in the presence of 0.1% BSA; (**B**). Brain mitochondria were isolated without BSA. Incubation conditions: 125 mM KCl, 10 mM MOPS, pH 7.2, 2 mM MgCl_2_, 2 mM KH_2_PO_4_, 10 mM NaCl, 1 mM EGTA, and 0.7 mM CaCl_2_, succinate 5 mM. Additions: ADP 150 µM, CCCP 0.5 µM, glutamate 10 mM (no malate). The release of SDH inhibition by glutamate did not depend on the presence of malate. In (**A**,**B**), respiratory rates are presented in nanomol O_2_/min/mg protein. The Figure was adapted from [11].

**Figure 2 ijms-25-12740-f002:**
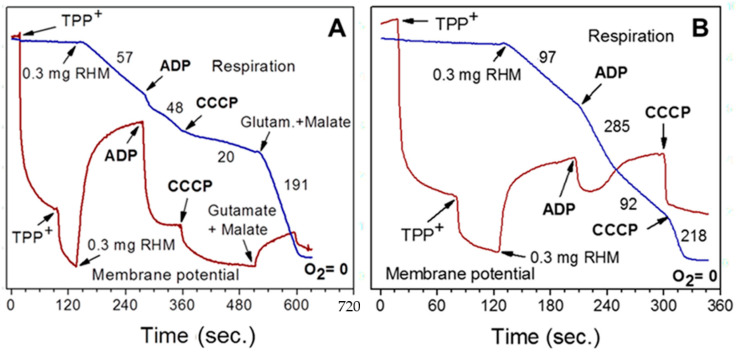
Oxidation of succinate and glutamate by isolated heart mitochondria from the Sprague Dawley rat. Incubation conditions and additions as in Figure 1. (**A**) rat heart mitochondria oxidizing succinate; (**B**) rat heart mitochondria oxidizing succinate + glutamate + malate. The Figure was adapted from [11].

**Figure 3 ijms-25-12740-f003:**
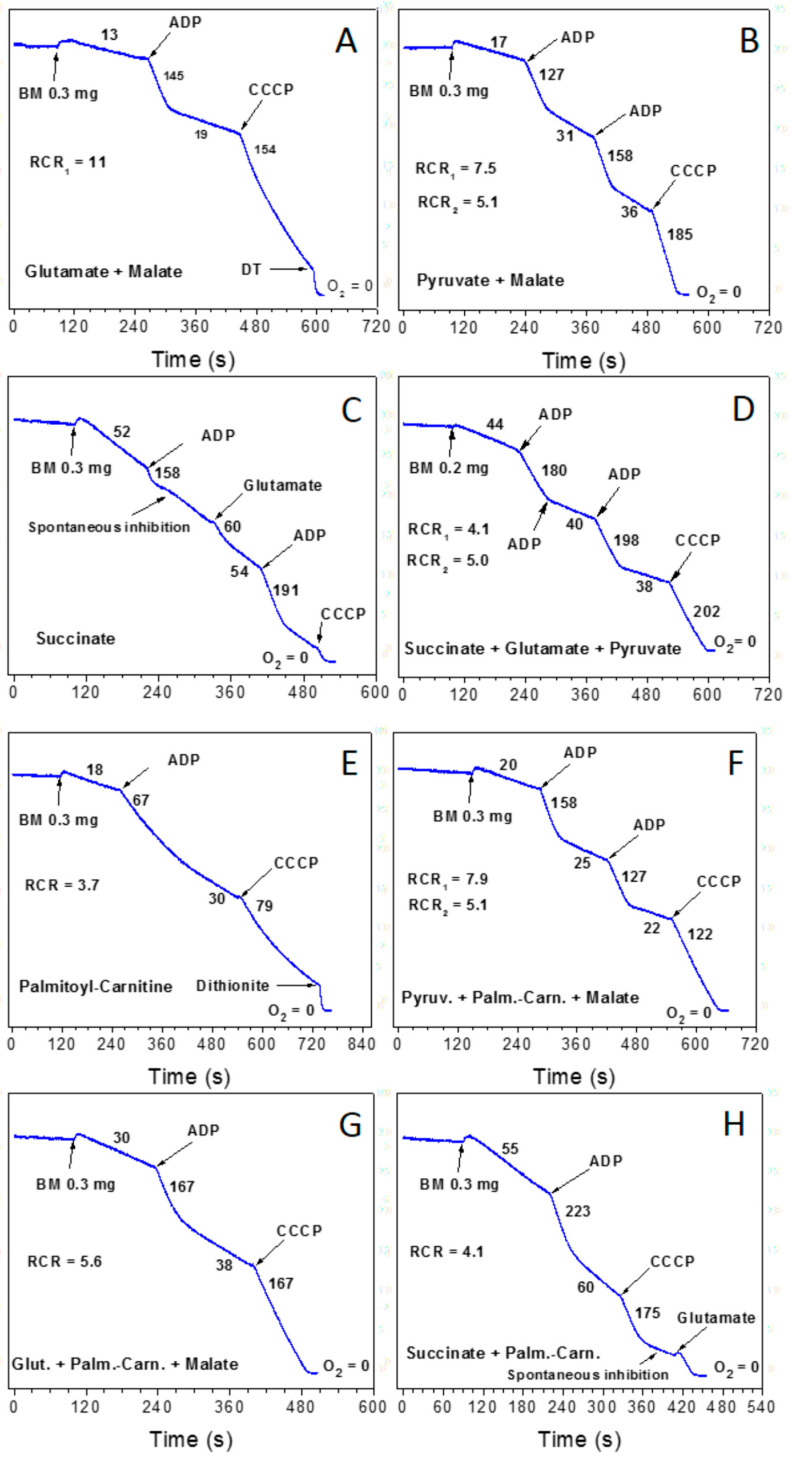
Oxygen consumption by rat (Sprague Dawley) brain mitochondria isolated without BSA oxidizing various substrates and their mixtures in different metabolic states. Substrates: Glutamate 5 mM, Malate 2 mM, Pyruvate 2.5 mM, Succinate 5 mM, Palmitoyl Carnitine 25 µM. Numbers at the traces are respiratory activities in nmol/min/mg mitochondrial protein. Respiratory activity ratio (RCR) is V_State 3_/V_State 4._ Additions: brain mitochondria 0.3 mg, ADP 150 µM, CCCP 0.5 µM, Glutamate 5 mM. The Figure was adapted from [12].

**Figure 4 ijms-25-12740-f004:**
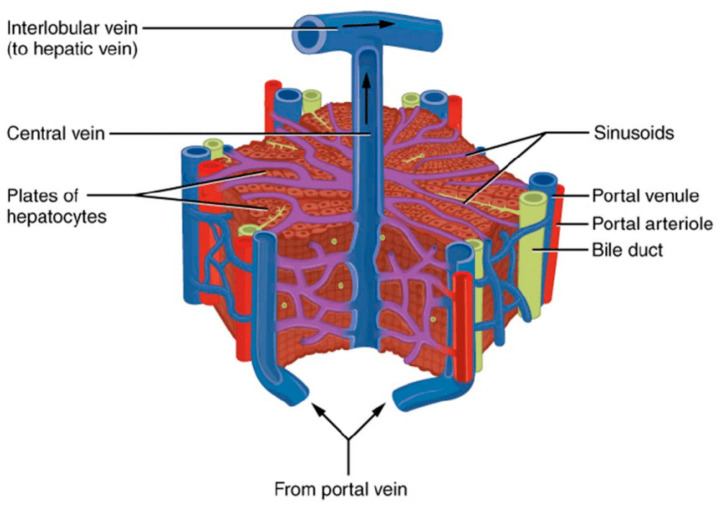
The hepatic lobule is the basic functional unit of the liver.

**Figure 5 ijms-25-12740-f005:**
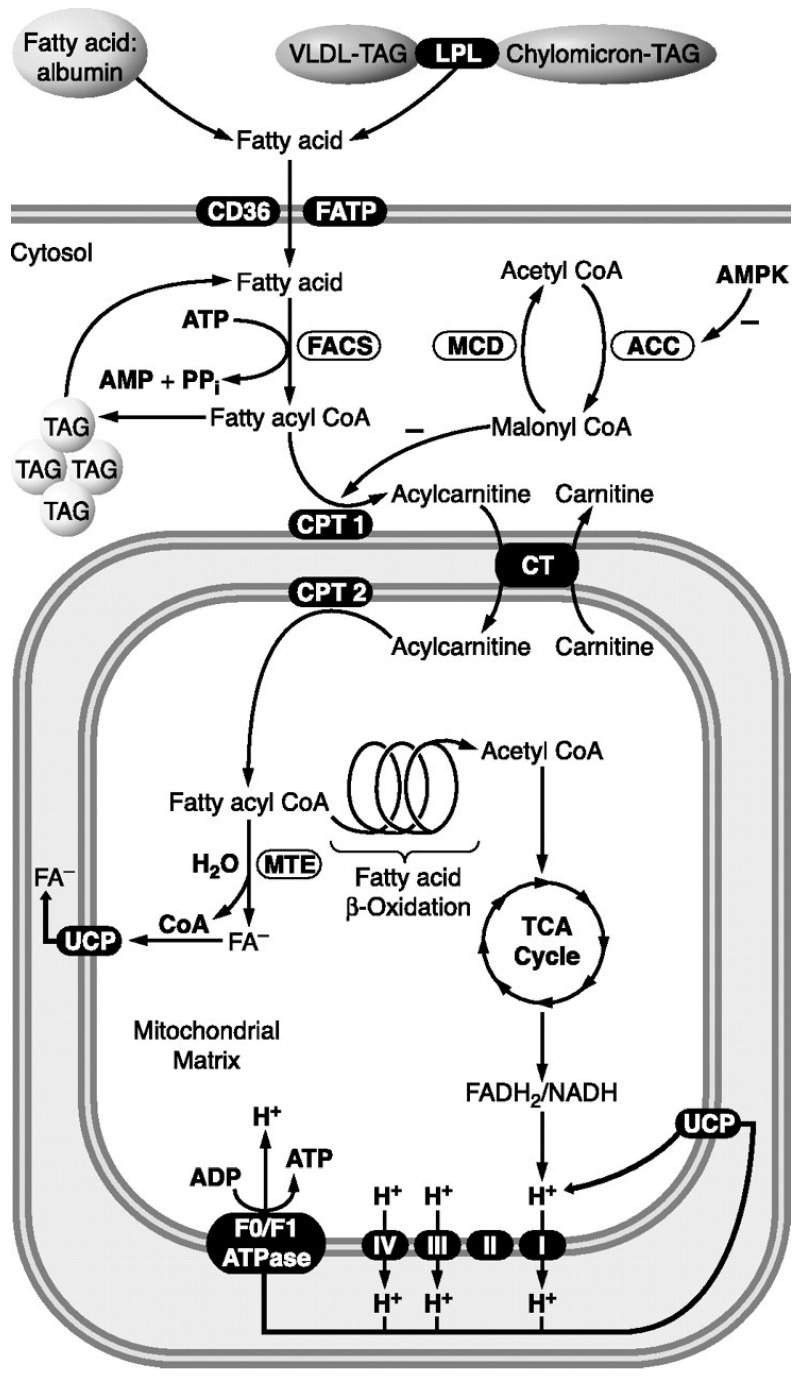
Overview of fatty acid β-oxidation in the heart.

**Figure 6 ijms-25-12740-f006:**
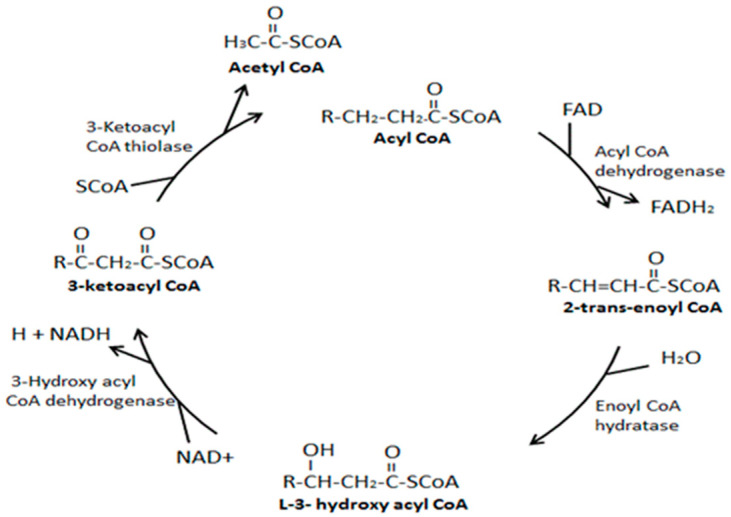
Old Paradigms mitochondrial fatty acids β-oxidation spiral. Adapted from [46].

**Figure 7 ijms-25-12740-f007:**
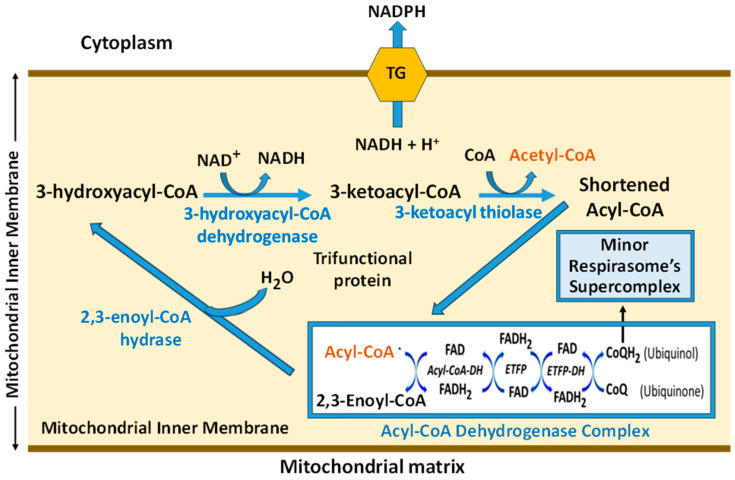
The multienzyme complexes of the cyclic β-oxidation of fatty acids. Abbreviations: Acyl-CoA DH—acyl-CoA dehydrogenase, ETPF—electron transfer protein, ETFP-DH—electron transfer protein dehydrogenase, TG—mitochondrial energy-dependent transhydrogenase.

**Figure 8 ijms-25-12740-f008:**
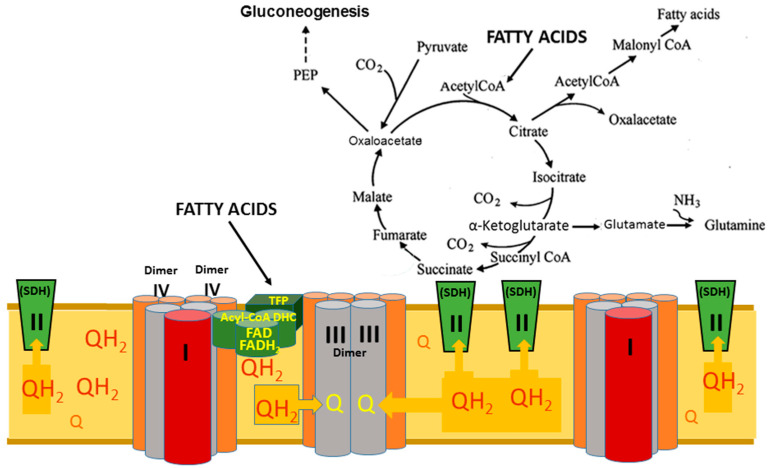
Functioning of respirasome and the Krebs Cycle during active β-oxidation of long-chain fatty acids. Abbreviations: Acyl-CoA DHC—acyl-CoA dehydrogenase complex, which includes three enzymes: acyl-CoA dehydrogenase, electron transfer flavoprotein (ETF), electron-transferring-flavoprotein dehydrogenase (ETFDH); PEP—phosphoenolpyruvate; TFP—trifunctional protein of the β-oxidation of the fatty acids system; SDH—succinate dehydrogenase; Q—ubiquinone, an oxidized form of coenzyme Q; QH_2_—ubiquinol, a reduced form of coenzyme Q. The figure was adapted from [8,12].

**Figure 9 ijms-25-12740-f009:**
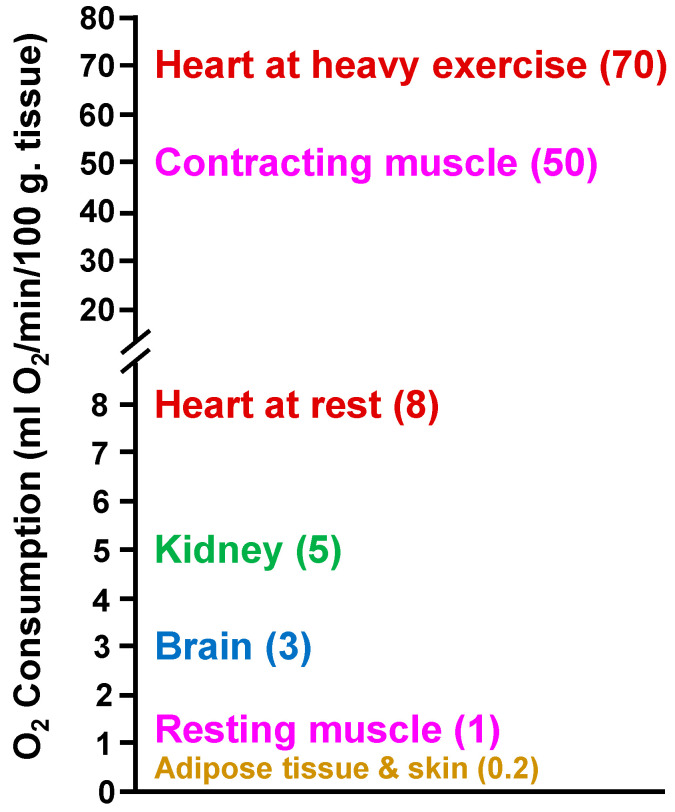
Tissue-specific oxygen consumption rates at various functional states. The figure was adapted from [68].

**Figure 10 ijms-25-12740-f010:**
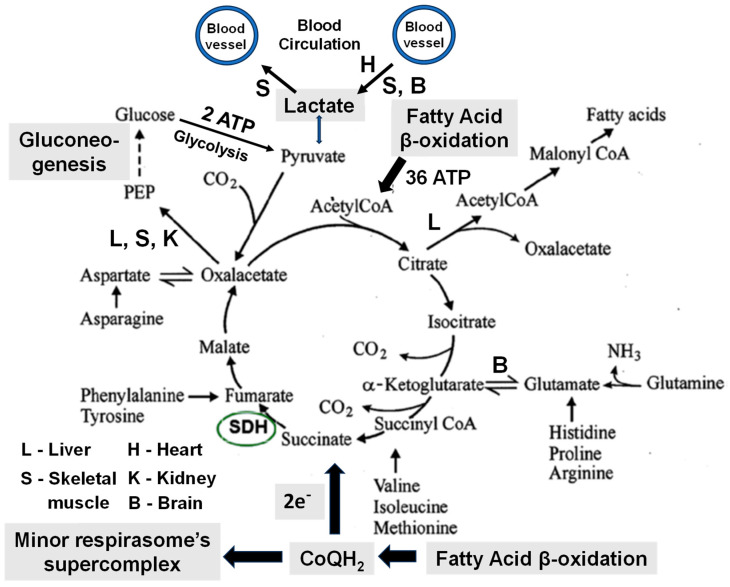
Schematic representation of various metabolic flow interactions aimed at maintaining ATP production in accordance with energy demands. Explanations in the text. The numbers in the arrows indicate the organs where this particular metabolic flow occurs.

**Table 1 ijms-25-12740-t001:** Caloric values, storage amount, and time of full storage consumption. The table was adapted from [15].

Source of Energy	Storage Amount (Time of Consumption)
**Carbohydrates:**	
Blood glucose and Glycogen in the liver	Total 4–5 g (20–30 min)
Caloric value = 3.81 kcal/g	100–120 g (1–3 h)
**Amino acids**	Released during catabolism of food, damaged tissue proteins, and anaplerotic pathways. The content is highly dynamic.
Caloric value = 3.12 kcal/g	
**Acyl Fatty Acids**	Fat storage kilograms, consumption duration in days
Caloric value = 9.3 kcal/g	

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
