# Peer review of "Role of Fatty Acids β-Oxidation in the Metabolic Interactions Between Organs"

_ijms, 2024, doi:10.3390/ijms252312740_

Round 1
Reviewer 1 Report
Comments and Suggestions for Authors
Panov et al. have written an excellent review article in which they explain the latest finding on energy production as it relates to fatty acid beta-oxidation. They explain that beta oxidation does not occur in a vacuum, but its intensity (required to meet the fuel requirements of oxidative-rich organs) is boosted to the cooperation of several different systems such as gluconeogenesis and lactate production. Furthermore, to provide the substrates in this process there needs to be a give and take between organs. These concepts are difficult to grasp and the authors do a great job of explaining them (with one exception noted below).
Major Comments:
In Section 3.2 and Figures 1 and 2, the authors are trying to explain (I think) simply that mitochondrial substrates such as succinate need help from other substrates like glutamate. This needs to be explained so much better and the figures need to be annotated in such a way that one can look at them and immediately see the differences and what they are talking about. After staring at Figure 1 for about 15 minutes, it was hard for me to see the differences. This may be because it took me 10 minutes to figure out what MBM meant. The authors need to define all the terms like MBM, tell what TPP+ does, explain how the membrane potential is relavent here, describe the states here (they do it after), define why the experiments are +/- BSA which is not even done until line 207 a few sections later. RCR is defined later also. Slopes are not quantified in Fig 1, but they are in Fig 2. Basically the authors need to do to Figure 1 (and to a lesser extent Fig 2) what they do to the rest of the paper which is to dumb it down for us mere mortals to understand. They also need to say why this section is even in this paper (is my assumption in the first sentence even correct?). Otherwise the rest of the paper reads like a dream, but we wouldn’t want the readers to get stuck on section 3.2 and put down the paper because the beauty is in the sections that follow.
Minor Comments
1. In Figure 2D, I think the BM (needs to be defined) should be 0.3 mg
2. Line 162-163 remind us that this is in brain
3. Line 324 …carnitine “is” oxidized…
4. Line 349 …reaction “to” determine…
5. Table 1, instead of writing CV (need to define if you use), you could fit the whole word and the units kcal/g right on the line so we don’t have to look it up
6. Line 434 …“but are” mainly produced…
7. Line 449 …signaling “mechanisms” have been….
8. I would label Fig 6: Old Paradigms of mitochondrial fatty acid….
9. There are some reference numbers messed up on lines 679,688
10. Figure 9, could you color code the organs so that it is easy to see the heart and muscle differences with exercise or something like that or even draw a line between them.
11. Figure 10, the numbers indicating organs are way too confusing to me (they get lost and confused with things like 2ATPs). Can you either write the whole word or use abbreviations like L for liver, Br for brain or something like that?
12. Throughout the paper and in the title the authors use the phrase Fatty Acids beta-Oxidation. I think the proper way to write this would be Fatty Acid beta-Oxidation without the S. I notice in some places they do write this, so maybe the editor’s call here.
Author Response
Round 1.
Comments and Suggestions for Authors
Panov et al. have written an excellent review article in which they explain the latest finding on energy production as it relates to fatty acid beta-oxidation. They explain that beta oxidation does not occur in a vacuum, but its intensity (required to meet the fuel requirements of oxidative-rich organs) is boosted to the cooperation of several different systems such as gluconeogenesis and lactate production. Furthermore, to provide the substrates in this process there needs to be a give and take between organs. These concepts are difficult to grasp and the authors do a great job of explaining them (with one exception noted below).
Authors' responses to the Reviewer’s comments. First, we express our gratitude to the reviewers for their job in the reviewing our manuscript and the valuable comments.
Reviewer 1. Major Comments:
In Section 3.2 and Figures 1 and 2, the authors are trying to explain (I think) simply that mitochondrial substrates such as succinate need help from other substrates like glutamate. This needs to be explained so much better and the figures need to be annotated in such a way that one can look at them and immediately see the differences and what they are talking about. After staring at Figure 1 for about 15 minutes, it was hard for me to see the differences. This may be because it took me 10 minutes to figure out what MBM meant. The authors need to define all the terms like MBM, tell what TPP+ does, explain how the membrane potential is relevant here, describe the states here (they do it after), define why the experiments are +/- BSA, which is not even done until line 207 a few sections later. RCR is defined later also. Slopes are not quantified in Fig 1, but they are in Fig 2. Basically the authors need to do to Figure 1 (and to a lesser extent Fig 2) what they do to the rest of the paper, which is to dumb it down for us mere mortals to understand. They also need to say why this section is even in this paper (is my assumption in the first sentence even correct?). Otherwise the rest of the paper reads like a dream, but we wouldn’t want the readers to get stuck on section 3.2 and put down the paper because the beauty is in the sections that follow.
Authors’ response. Thank you very much for noticing that we did not explain the significance of Figures 1 and 2 to the readers. This is a very important admonition. We introduced into section 3.2. the following introduction: “3.2. Endogenous inhibition of succinate dehydrogenase (SDH). Activation of fatty acids β-oxidation, which occurs in the presence of supporting substrates [6], is associated with the reversal of the succinate dehydrogenase (SDH) reaction when the electrons flow from the membrane’s pool of CoQH2 to the tricarboxylic acid cycle [discussed in 10]. Thus, SDH (complex II) plays a critical role in fatty acid beta-oxidation by reversing electron flow and preventing catabolism of mitochondrial metabolites. Therefore, it is important to discuss another phenomenon observed in the in vitro experiments with succinate as a sole substrate: the endogenous inhibition of SDH caused by oxaloacetate”.
As regards the abbreviations, we apologize for missing them. We corrected these omissions. Thank you.
Minor Comments
- In Figure 2D, I think the BM (needs to be defined) should be 0.3 mg.
Authors’ response. Yes, it should be 0.3 mg of mitochondrial protein. However, since I tried to do this set of experiments on the same batch of isolated mitochondria in this particular experiment, which was the last one, I had to use 0.2 mg. I am trying to be honest regarding experiments.
- Line 162-163 remind us that this is in brain
Authors’ response. I have corrected the sentence: “Unfortunately, isolated mitochondria from all organs in the in vitro system oxidize the carnitine esters of the long-chain fatty acids very poorly, including the brain mitochondria (Fig.3E)”.
- Line 324 …carnitine “is” oxidized
Authors’ response. Corrected
- Line 349 …reaction “to” determine…
Authors’ response. Corrected.
- Table 1, instead of writing CV (need to define if you use), you could fit the whole word and the units kcal/g right on the line so we don’t have to look it up
Authors’ response. I followed your advice. Indeed, it looks better.
- Line 434 …“but are” mainly produced…
Authors’ response. Corrected.
- Line 449 …signaling “mechanisms” have been….
Authors’ response. Corrected.
- I would label Fig 6: Old Paradigms of mitochondrial fatty acid….
Authors’ response. A good advice. I changed the label for Fig.6.
- There are some reference numbers messed up on lines 679,688.
Authors’ response. Corrected.
- Figure 9, could you color code the organs so that it is easy to see the heart and muscle differences with exercise or something like that, or even draw a line between them?
Authors’ response. We followed your advice and color-coded organs in Fig.9.
- Figure 10, the numbers indicating organs are way too confusing to me (they get lost and confused with things like 2ATPs). Can you either write the whole word or use abbreviations like L for liver, Br for brain or something like that?
Authors’ response. We followed your advice and in Figure 10 indicated organs with initial letters in bold.
- Throughout the paper and in the title the authors use the phrase Fatty Acids beta-Oxidation. I think the proper way to write this would be Fatty Acid beta-Oxidation without the S. I notice in some places they do write this, so maybe the editor’s call here.
Authors’ response. I agree that the singular is usually used in this metabolic pathway name. However, the acyl-CoA dehydrogenase complex comprises four enzymes specific for fatty acids with different chain lengths. So, beta-oxidation catabolizes the simultaneous catabolism of different FAs. To emphasize this, we used the plural.
Submission Date
25 October 2024
Date of this review
05 Nov 2024 22:14:26
Round 2.
Comments and Suggestions for Authors
I have gone through the whole review article under the title “Role of Fatty Acids β-Oxidation in the Metabolic Interactions Between Organs” and explained the special role of fatty acids regarding ATP production and their alteration by effecting the glycolysis pathway in various ways. I found that scientifically, the manuscript is written very well. However, the following are some minor mistakes which need corrections.
- In the whole manuscript such as line 10, 11, 13 and 17, 19, 20, 31, 34, 48, 50, and 129 etc. there are some spelling mistakes please double check that.
Authors’ response. First of all, we express our gratitude to the Reviewer for the valuable comments and suggestions. We have double-checked the text for misspellings and grammar.
- Please write the scientific terms in italic like “In-vitro” found in line 321.
Authors’ response. Corrected.
- In Table 1, you have pointed out (CV = 3.12) with Asterisk “*”, what do you want to show? Please elaborate the meaning of Asterisk below the table.
Authors’ response.
- In line 500 and 535, no citation or reference was found please review and add reference.
Authors’ response. “*” has no meaning and was removed.
- In line 512, you have written (unpublished data) so please elaborate this with a reference, it will be better.
Authors’ response. The observation that human blood samples quickly lose oxygen was made by accident and never published.
- In line 679, please correct the citation according to the journal’s format, same in line 688.
Authors’ response. Corrected.
- It is suggested to elaborate more the sentence “it was an important discovery that in vivo, 770 the product of glycolysis is always lactate, not pyruvate” (line 770, 771 and 772) with some detail.
Authors’ response. Thank you for this valuable suggestion. We have added explanatory sentences, as shown in blue. Traditionally, the end product of glycolysis has been viewed as dependent on tissue oxygenation: pyruvate in the presence of adequate oxygenation and lactate - in the absence of proper oxygenation. Rogatzki et al. [100] considered differences in the enzyme activities, namely that the lactate dehydrogenase (LDH) reaction is at near-equilibrium and has much higher activity than the regulatory enzymes of the glycolytic and oxidative pathways, as the main reason that lactate is always the end product of glycolysis. According to our hypothesis, presented here, there are two reasons why in vivo glycolysis ends up in the formation of lactate: 1) fatty acid β-oxidation generates a high NADH/NAD+ ratio, and 2) glycolytic pyruvate is unable to enter the TCA cycle. All these facts were impossible to understand and explain based on classical enzymology and thermodynamics paradigms.
Submission Date
25 October 2024
Date of this review
13 Nov 2024 03:40:26
© 1996-2024 MDPI (Basel, Switzerland) unless otherwise stated
Reviewer 2 Report
Comments and Suggestions for Authors
I have gone through the whole review article under the title “Role of Fatty Acids β-Oxidation in the Metabolic Interactions Between Organs” and explained the special role of fatty acids regarding ATP production and their alteration by effecting the glycolysis pathway in various ways. I found that scientifically, the manuscript is written very well. However, the following are some minor mistakes which need corrections.
· In the whole manuscript such as line 10, 11, 13 and 17, 19, 20, 31, 34, 48, 50, and 129 etc. there are some spelling mistakes please double check that.
· Please write the scientific terms in italic like “In-vitro” found in line 321.
· In Table 1, you have pointed out (CV = 3.12) with Asterisk “*”, what do you want to show? Please elaborate the meaning of Asterisk below the table.
· In line 500 and 535, no citation or reference was found please review and add reference.
· In line 512, you have written (unpublished data) so please elaborate this with a reference, it will be better.
· In line 679, please correct the citation according to the journal’s format, same in line 688.
· It is suggested to elaborate more the sentence “it was an important discovery that in vivo, 770 the product of glycolysis is always lactate, not pyruvate” (line 770, 771 and 772) with some detail.
Author Response
Round 2.
Comments and Suggestions for Authors
I have gone through the whole review article under the title “Role of Fatty Acids β-Oxidation in the Metabolic Interactions Between Organs” and explained the special role of fatty acids regarding ATP production and their alteration by effecting the glycolysis pathway in various ways. I found that scientifically, the manuscript is written very well. However, the following are some minor mistakes which need corrections.
- In the whole manuscript such as line 10, 11, 13 and 17, 19, 20, 31, 34, 48, 50, and 129 etc. there are some spelling mistakes please double check that.
Authors’ response. First of all, we express our gratitude to the Reviewer for the valuable comments and suggestions. We have double-checked the text for misspellings and grammar.
- Please write the scientific terms in italic like “In-vitro” found in line 321.
Authors’ response. Corrected.
- In Table 1, you have pointed out (CV = 3.12) with Asterisk “*”, what do you want to show? Please elaborate the meaning of Asterisk below the table.
Authors’ response.
- In line 500 and 535, no citation or reference was found please review and add reference.
Authors’ response. “*” has no meaning and was removed.
- In line 512, you have written (unpublished data) so please elaborate this with a reference, it will be better.
Authors’ response. The observation that human blood samples quickly lose oxygen was made by accident and never published.
- In line 679, please correct the citation according to the journal’s format, same in line 688.
Authors’ response. Corrected.
- It is suggested to elaborate more the sentence “it was an important discovery that in vivo, 770 the product of glycolysis is always lactate, not pyruvate” (line 770, 771 and 772) with some detail.
Authors’ response. Thank you for this valuable suggestion. We have added explanatory sentences, as shown in blue. Traditionally, the end product of glycolysis has been viewed as dependent on tissue oxygenation: pyruvate in the presence of adequate oxygenation and lactate - in the absence of proper oxygenation. Rogatzki et al. [100] considered differences in the enzyme activities, namely that the lactate dehydrogenase (LDH) reaction is at near-equilibrium and has much higher activity than the regulatory enzymes of the glycolytic and oxidative pathways, as the main reason that lactate is always the end product of glycolysis. According to our hypothesis, presented here, there are two reasons why in vivo glycolysis ends up in the formation of lactate: 1) fatty acid β-oxidation generates a high NADH/NAD+ ratio, and 2) glycolytic pyruvate is unable to enter the TCA cycle. All these facts were impossible to understand and explain based on classical enzymology and thermodynamics paradigms.
Submission Date
25 October 2024
Date of this review
13 Nov 2024 03:40:26
© 1996-2024 MDPI (Basel, Switzerland) unless otherwise stated
Round 2
Reviewer 2 Report
Comments and Suggestions for Authors
The article has been greatly improved, it is recommended to accept the article